# CENPE Inhibition Leads to Mitotic Catastrophe and DNA Damage in Medulloblastoma Cells

**DOI:** 10.3390/cancers13051028

**Published:** 2021-03-01

**Authors:** Giorgia Iegiani, Marta Gai, Ferdinando Di Cunto, Gianmarco Pallavicini

**Affiliations:** 1Neuroscience Institute Cavalieri Ottolenghi, 10043 Turin, Italy; giorgia.iegiani@unito.it; 2Department of Neuroscience ‘Rita Levi Montalcini’, University of Turin, 10126 Turin, Italy; 3Department of Molecular Biotechnology and Health Sciences, University of Turin, 10126 Turin, Italy; marta.gai@unito.it

**Keywords:** CENPE, microcephaly, DNA damage, childhood brain tumor, 53BP1, γH2AX, mitotic catastrophe

## Abstract

**Simple Summary:**

Medulloblastoma (MB) is the most frequent brain tumor in children. The standard treatment consists in surgery, followed by radiotherapy and chemotherapy. These therapies are only partially effective, since many patients still die and those who survive suffer from neurological and endocrine disorders. Therefore, more effective therapies are needed. CENPE is a gene critical for normal proliferation and survival of neural progenitors. Since there is evidence that MB cells are very similar to neural progenitors, we hypothesized that CENPE could be an effective target for MB treatment. In MB cell lines, CENPE depletion induced defects in division and resulted in cell death. To consolidate CENPE as a target for MB treatment, we tested GSK923295, a specific inhibitor already in clinical trials for other cancer types. GSK923295 induced effects similar to CENPE depletion at low nM levels, supporting the idea that CENPE’s inhibition could be a viable strategy for MB treatment.

**Abstract:**

Medulloblastoma (MB) is the most frequent brain tumor in children. The standard treatment consists in surgery, followed by radiotherapy and chemotherapy. These therapies are only partially effective since many patients still die and those who survive suffer from neurological and endocrine disorders. Therefore, more effective therapies are needed. Primary microcephaly (MCPH) is a rare disorder caused by mutations in 25 different genes. Centromere-associated protein E (CENPE) heterozygous mutations cause the MCPH13 syndrome. As for other MCPH genes, CENPE is required for normal proliferation and survival of neural progenitors. Since there is evidence that MB shares many molecular features with neural progenitors, we hypothesized that CENPE could be an effective target for MB treatment. In ONS-76 and DAOY cells, CENPE knockdown induced mitotic defects and apoptosis. Moreover, CENPE depletion induced endogenous DNA damage accumulation, activating TP53 or TP73 as well as cell death signaling pathways. To consolidate CENPE as a target for MB treatment, we tested GSK923295, an allosteric inhibitor already in clinical trial for other cancer types. GSK923295, induced effects similar to CENPE depletion with higher penetrance, at low nM levels, suggesting that CENPE’s inhibition could be a therapeutic strategy for MB treatment.

## 1. Introduction

Medulloblastoma (MB) is the most common malignant brain tumor in children [1,2]. MB has been classified into four biological subgroups based on microarray and genomic sequencing technologies: WNT, Sonic Hedgehog (SHH), Group 3 and Group 4 [3,4,5,6,7]. MB standard treatment is surgery, followed by irradiation of the entire neuro-axis and high-dose multi-agent chemotherapy [2]. Despite long-term survival can be as high as 90% in the rare WNT subgroup, it is on average around 50% in the other subtypes, with an intermediate prognosis in Group 4 and worse in Group 3 patients [8]. Despite optimal treatment, many patients still die and those who survive suffer very often from neurological, cognitive and endocrine disorders, caused by the aggressive therapy [8].

For all these reasons, more effective therapies are necessary. Targeting driver mutations is a common strategy to develop new anti-cancer drugs. For example, in the SHH subgroup various SHH pathway inhibitors, including vismodegib and erismodegib, have been tested. [9]. Unfortunately, few patients responded and resistance to the treatment was frequently observed [10,11,12].

An alternative drug target is represented by the large class non-oncogene proteins which, despite not being mutated, are nevertheless required for tumor growth and progression [13]. Although the cellular origin of the different MB subtypes is still debated, it is clear that MB cells share many molecular features with cerebellar granule progenitors and radial glia cells [14,15]. On this basis, genes that are selectively required during development for proliferation and genomic stability of neural progenitors [16] but not for other central nervous system cell types, may represent attractive targets for MB-directed drug development.

On this ground, genes mutated in primary microcephaly (MCPH) syndromes have already been proposed as possible targets for MB-directed drug development [17,18,19]. 

MCPH is a rare and heterogeneous disease characterized by reduced brain volume as compared to the rest of the body [20]. 25 MCPH genes have been identified [21,22]. Many of these genes encode for proteins associated to centrosomes and their loss leads to cell cycle delays and spindle mis-orientation [23]. Consequently, the altered balance between symmetric and asymmetric division may cause a reduction of proliferating progenitors either through apoptosis or premature differentiation [23,24]. Moreover, the loss of various MCPH genes leads to DNA damage accumulation [24,25,26,27,28,29]. 

Heterozygous mutations of centromere-associated protein E (CENPE) cause the MCPH13 syndrome, leading to a reduction of head circumference of 7–9 standard deviation below the mean [30]. CENPE is a microtubule plus-end-directed kinetochore motor protein, important in chromosome congression, spindle microtubule capture at kinetochores and spindle assembly checkpoint (SAC) [31,32,33,34]. It is expressed during cell cycle, reaching its peak during G2/M phase [35]. During the transition to anaphase, CENPE translocates to the central spindle and is then localized at the central region of midbody during cytokinesis [36]. Disruption of CENPE function prevents chromosome alignment, inhibits microtubules attachment to kinetochores and induces mitotic arrest [37,38] in prometaphase and metaphase [39]. CENPE-depleted cells display also a long delay in the metaphase-to-anaphase transition [39] even though no effects are observed on mitotic spindle assembly [40]. Mitotic slippage of CENPE-deficient cells, obtained by relieving the SAC, leads to chromosome mis-segregation, proteotoxic stress, DNA damage and P53-dependent apoptosis [41].

CENPE has been associated to tumorigenesis in various types of cancer [42,43,44,45] although few studies have been conducted in brain tumors [46]. The ATP-dependent microtubule binding domain is essential for CENPE localization at mitotic spindles [47]. Moreover, the ATPase function promotes microtubule elongation at kinetochore and stabilize its conformation [48]. Different CENPE inhibitors have been developed [41,49]. The most attention-grabbing is GSK923295, an allosteric inhibitor that binds the ATPase pocket [49] and shows broad in vivo antitumor activity in preclinical models [50,51]. Phase I clinical studies, performed in adult patients with solid tumors non responsive to common therapies, demonstrated a dose-proportional pharmacokinetic with mild adverse effects [52].

In this report we analyzed the possibility that CENPE depletion or inhibition may exert strong anti-proliferative effects in MB cells. Our results support the idea that CENPE is a promising target for this tumor type.

## 2. Materials and Methods

### 2.1. Cell Culture

DAOY cells were obtained from ATCC and were cultured in MEM medium (Euroclone, Milan, Italy) supplemented with 10% fetal bovine serum (FBS, Gibco, Gaithersburg, MD, USA), nonessential amino acids (Gibco), L-glutamine (Gibco), sodium pyruvate (Gibco) and 1% penicillin/streptomycin (Life Technologies, ThermoFisher, Waltham, MA, USA). ONS-76 cells were kindly provided by Luigi Varesio (Gaslini Hospital, Genova, Italy) and were cultured in RPMI medium supplemented with 10% FBS and 1% penicillin/streptomycin (Life Technologies, ThermoFisher). For both cell lines, the number of in vitro passages from thawing of the original aliquots to experiments was comprised between 5 and 8. Cells were routinely analysed for morphological features and tested for Mycoplasma contamination with the following oligonucleotides sequences: MYCO1: 5′-ACTCCTACGGGAGGCAGCAGTA-3′; MYCO2: 5′-TGCACCATCTGTCACTCTGTTAACCTC-3′.

### 2.2. Transfection and RNAi

For knockdown of CENPE ONTARGETplus Dharmacon SMART pools were used (Dharmacon, Lafayette, CO, USA). D-001810-10 non-targeting pool was used as a non-targeting control. ONS-76 and DAOY cells plated on six-well plates were transfected using 6.25 μL of the required siRNA (20 μM) together with 1.5 μL Lipofectamine 2000 (Invitrogen, Carlsbad, CA, USA), according to the manufacturer’s instructions. Efficient knockdown was obtained after 48 h.

### 2.3. Drug Treatment and EC50 Calculation

CENPE activity was inhibited by GSK923295. To calculate DAOY and ONS-76 EC50 values, cells were plated in triplicates into a 12-well plate and the following concentration were used: 1 nM, 10 nM, 25 nM, 50 nM, 100 nM and 1000 nM. After 24, 48 and 72 h cells were counted with Bio-Rad TC20 Automated Cell Counter (Bio-Rad, Hercules, CA, USA) and the values were fitted into a nonlinear regression curve on GraphPad 8 (GraphPad Software, San Diego, CA, USA).

### 2.4. Analysis of Cell Proliferation

To analyze cell proliferation, 5 × 10^4^ DAOY and ONS cells were seeded into 12-well plate, transfected with siCtrl or siCENPE and counted in triplicates after 48, 72 and 96 h with Bio-Rad TC20 Automated Cell Counter. To assess the effects of GSK923295, 5 × 10^4^ DAOY and ONS cells were seeded into 12-well plate, treated with DMSO or 25 nM GSK923295 and counted in triplicates after 24, 48 and 72 h with Bio-Rad TC20 Automated Cell Counter.

### 2.5. Colony Forming Assay 

For colony forming assay, 300 cells were plated at day 1 for DAOY and ONS-76 in medium containing DMSO or 25 nM GSK923295. The clonogenic assay was stopped after 7 days. To assess the phenotype of cell resistant to treatment we plated 10000 cells. After medium removal, colonies were stained for 10 min with Nissl staining (0.1% cresyl violet acetate and 0.6% glacial acetic acid) and rinsed in water. 

### 2.6. CellTox™ Green Cytotoxicity Assay 

Accordingly to the manufacturer’s instructions, 20 µL of concentrated CellTox™ Green Reagent (Promega, Madison, WI, USA) were added to each well at the end point of treatment. The plate was mixed on an orbital shaker (700–900 rpm) for 1 min to ensure homogeneity and incubated at room temperature for 15 min, shielded from ambient light. Finally, we measured fluorescence intensity at 485–500 nmEx/520–530 nmEm.

### 2.7. Antibodies 

The following antibodies were used: rabbit polyclonal anti-CENPE (#C7488 Sigma-Aldrich, Saint Louis, MO, USA) mouse monoclonal mouse monoclonal anti-vinculin (#V9131 Sigma-Aldrich); rabbit polyclonal anti-γH2AX (S139) (20E3) (#2577 Cell Signaling Technology, Danvers, MA, USA), rabbit polyclonal anti-cleaved Caspase 3 (#9661S Cell Signaling Technology), rabbit polyclonal anti-P21 (#sc-756 SantaCruz, Dallas, TX, USA), rabbit polyclonal anti-TP53 phospho Ser15 (#9284S Cell Signaling Technology), mouse monoclonal anti-TP53 (1C12) (#2524S Cell Signaling Technology), rabbit polyclonal anti-TP73 (#ab14430 Abcam, Cambridge, UK), mouse monoclonal anti-αTubulin (#T5168 Sigma-Aldrich).

### 2.8. Western Blotting

Cell lines were lysed in RIPA buffer (1% NP40, 150 mM NaCl, 50 mM TRIS HCl pH 8.5 mM EDTA, 0.01% SDS, 0.005% sodium deoxycholate, Roche protease inhibitors, PMSF) for 10 min at 4 °C. Samples were clarified 10 min at 13,000 rpm at 4 °C. For immunoblots, equal amounts of proteins from both whole-cell lysates were resolved by SDS–PAGE and blotted to nitrocellulose membranes. 

### 2.9. Immunofluorescence and Live Imaging

Cultured cells were fixed 5 min at RT using PFA 2% then treated 10 min at RT using CSK buffer [100 mM NaCl, 300 mM sucrose, 3 mM MgCl_2_, 10 mM PIPES (pH 6.8), 0.7% Triton], and finally fixed again 5 min at RT using PFA 2%. Cells were permeabilized in 0.1% Triton X-100 in PBS for 10 min, saturated in 5% BSA in PBS for 30 min and incubated with a primary antibody for 1 h at RT. Primary antibodies were detected with anti-rabbit Alexa Fluor 488 or 555 (Molecular Probes, Invitrogen, Eugene, OR, USA), anti-mouse Alexa Fluor 488 or 555 (Molecular Probes, Invitrogen) used at 1:10000 dilution for 30 min. Cells were counterstained with 0.5 μg/mL DAPI for 10 min and washed with PBS. Images were acquired using a Leica TCS SP5 confocal system (Leica Microsystems, Wetzlar, Germany) equipped with a 405-nm diode, an argon ion, and a 555-nm DPSS laser. For live imaging, time lapses were recorded overnight with an interval of 5 min using a 40× PlanApo N.A. 1.4 oil immersion objective on the cells kept in the microscope incubator at 37 °C and 5% CO_2_. Time lapses were manually analyzed with ImageJ software (Version 2.1.0/1.53c, Rasband, W.S., U.S. National Institutes of Health, Bethesda, MD, USA). Metaphase length was measured from the first image in which a round mitotic cell is clearly visible to anaphase onset; cell death during metaphase (before anaphase onset) was classified as mitotic catastrophe; cells that fuse back together after anaphase were classified as cytokinesis failures.

### 2.10. Survival Analysis 

CENPE expression in human patients with medulloblastoma was downloaded from Gene Expression Omnibus (GEO) dataset GSE85218 [53], generated on the Affymetrix Human Gene 1.1 ST Array platform (https://www.ncbi.nlm.nih.gov/geo/query/acc.cgi?acc=GSE85218, accessed on 15 February 2021). Data are shown as RSEM count. 

Overall Survival (OS) data were obtained from [53] and maximal survival time was cut at 10 years; patients without survival data were excluded. Patients were divided in two groups, “low CENPE” and “high CENPE” using median expression as the discriminant value. Log-rank (Mantel-Cox) test was used to compare survival between groups. 

### 2.11. Statistical Analysis 

Statistical analyses were performed using Microsoft Office Excel (Version 16, Microsoft Corporation, Redmond, WA, USA) and GraphPad (Version 8, GraphPad Software, San Diego, CA, USA). Unpaired two-tails Student’s *t*-test was used if not otherwise specified. Data are shown as the mean values of at least 3 independent experiments and standard error of the mean (mean ± SEM). Mann-Whitney test was used to analyze 53BP1 foci and Chi^2^ for percentage distributions using absolute frequency of experiments. 

## 3. Results

### 3.1. Medulloblastoma Cells Are Strongly Sensitive to CENPE Knockdown

To investigate the dependence of MB on CENPE expression, we resorted to gene expression data across 496 primary MB samples analyzed in [53]. Kaplan-Meier survival curves of MB patients were obtained dividing patients between high and low CENPE expression groups, defined on the basis of CENPE probe sets expression median value (Figure 1A). As expected, low levels of CENPE correlates with a better prognosis. Expression levels were not statistically different between MBs subgroups (Figure 1B). Interestingly, Kaplan-Meier survival curves revealed a significant correlation between CENPE low levels and longer survival only in the SHH subgroup (Figure 1C). This result is consistent with SHH MBs arising from cerebellar granule neuron progenitors [54], which are the cells most affected by CENPE loss during neuro-development [30].

In light of these results, to validate CENPE as promising target for MB, we selected DAOY and ONS-76 cells, two established human MB lines showing a transcriptional signature of SHH activation [1] and carrying mutated or wild type TP53, respectively [55]. We induced transient CENPE knockdown in these cells (Figure 2A), using a pool of specific siRNAs, and analyzed cellular phenotypes 48 h after transfection. As expected [40], CENPE knockdown did not alter mitotic spindle assembly (Figure 2B). Nevertheless, live cell imaging showed clear mitotic defects (Figure 2C–E). In particular, as previously described in HeLa, mitotic cells of both lines were strongly delayed in their metaphase-to-anaphase transition (Figure 2C,D; see also Appendix A). After CENPE depletion, DAOY and ONS-76 showed prolonged metaphase in 50% and 40% of divisions, with and average duration of 66 and 36 min, respectively (Figure 2D,E). Interestingly, DAOY cells were dramatically affected, since 17% of divisions ended up with mitotic catastrophe before completing cytokinesis. Consistent with the observed mitotic defects, CENPE knockdown strongly impaired the expansion of DAOY and ONS-76 cells, if compared to controls (Figure 2F,G).

### 3.2. CENPE Knockdown Induces Apoptosis and DNA Damage in MB Cells

The mitotic catastrophes in DAOY cells and the strongly reduced growth rate of both cell lines, observed after CENPE knockdown, suggest that MB cells may require CENPE not only for normal proliferation but also for viability. To confirm the induction of cell death, we resorted to CellTox™ Green Cytotoxicity assay [56], which measures change in membrane integrity. 48 h after treatment with CENPE-directed siRNAs, a strong increase in cell death was detected in DAOY cells. A lower but significant increase was also observed in ONS-76 cells (Figure 3A). In agreement with these observations, western blot analysis revealed a significant increase of cleaved Caspase 3 in both cell lines, suggesting the induction of apoptosis (Figure 3B,C). In HeLa cells, CENPE inactivation leads to apoptosis and accumulation of DNA damage when the spindle checkpoint is concomitantly inactivated [41]. Since MB cells share many aspects of their expression profile with neural progenitors, we reasoned that they might be more sensitive than HeLa cells to CENPE knockdown. We therefore asked whether CENPE inactivation may also lead to increased DNA damage, even without other SAC components inhibition. Western blot analysis revealed a significant increase in γH2AX, a marker of both single and double strand brakes (Figure 3B,D). We also quantified the frequency of nuclear foci of 53BP1 protein, a specific marker of DNA-double strand breaks [57,58] 48 h after CENPE knockdown. This analysis revealed a significant increase both in DAOY and ONS-76 cells (Figure 3E,F). Altogether, these results indicate that CENPE inactivation is sufficient to produce DNA damage and apoptosis in MB cells.

### 3.3. CENPE Inhibition Reduces Proliferation and Induces Cell Death in MB Lines

Based on the encouraging results obtained with transient depletion, we addressed the effect of the highly selective CENPE inhibitor GSK923295 [49] on SHH MB cells. In particular, we generated dose/response curves at different times (Figure 4A,B). GSK923295 reduced the expansion of both DAOY and ONS-76 cells, with an EC_50_ of 12 nM and 14 nM at 72 h, respectively. Using a CellTox™ Green Cytotoxicity assay, we determined dose/response of DAOY and ONS-76 to GSK923295 at 24 h in terms of cell death (Figure 4C,D). In this case, the EC_50_ was 18 nM for DAOY and 178 nM for ONS-76. Finally, using the lower effective dose on both cell lines at 24 h post treatment (25 nM), we tested the effects of GSK923295 in long term proliferation assay. Proliferation curves highlighted that, during the first days of treatment, DAOY and ONS-76 were deeply affected, but with different penetrance (Figure 4E,F). Despite this, GSK923295 completely abolished the in vitro clonogenic potential of both cell lines after 7 days of treatment, as revealed by colony forming assay (Figure 5A,B). To evaluate the proportion of cells that resist to the treatment, we plated ten thousand cells with 25 nM of GSK923295 in a six well plate for 7 days. In every plate, single cells or abortive colonies (less than 10 cell per colony) were detected. Among them, 86% and 73%, respectively, were composed of single cells, characterized by a flattened senescent morphology [59] (Figure 5C,D). These results confirm that SHH MB cells are highly sensitive to CENPE inhibition and that DAOY cells are more susceptible than ONS-76 cells.

### 3.4. CENPE Inhibition Alters Mitotic Spindle Assembly and Induces Mitotic Catastrophe in MB Cells

Analysis of mitotic cells by immunofluorescence revealed that GSK923295 had stronger effects than CENPE knockdown. Indeed, although treated cells were still capable of assembling bipolar spindles, these were more disorganized and less focused at their poles. The signal of CENPE was not evenly co-localized with spindle microtubules, but the protein was strongly enriched at the spindle poles; moreover, protein levels appeared to be increased, suggesting that the catalytic activity may affect protein production or turnover (Figure 6A). In addition, consistent with previous data, not all chromosomes were congressed at the midzone at metaphase, but some of them remained closely associated with the spindle poles [60]. Live cell imaging showed mitotic phenotypes similar to those produced by CENPE knockdown, but with higher penetrance (Figure 6B–E; see Appendix A). In particular, both DAOY and ONS-76 cells showed a higher percentage of cytokinesis failures. Approximately 80% of DAOY mitosis ended up with mitotic catastrophe and such events became detectable even in ONS-76 cells (Figure 6E). In agreement with these data, biochemical analysis of cells treated for 24 h with the inhibitor reveled increased levels of cleaved Caspase 3 and γH2AX (Figure 6F–H).

Taken together, these observations indicate that CENPE inhibition disrupts mitosis and induces cytokinesis failure and mitotic catastrophe in both MB cell lines, with higher penetrance in DAOY cells. Moreover, the increased levels of DNA damage marker H2AX suggest that apoptosis and growth arrest could be the result of genotoxic stress. Accordingly, ONS-76 cells treated with CENPE-specific siRNA showed TP53 and P21 activation, which were even more pronounced after GSK923295 treatment (Figure 7A–D). Increased P21 levels were detected also in DAOY cells, which express a mutated form of P53 (Figure 7E,F). Interestingly, the latter cells showed induction of the TP53 family member TP73, after both CENPE knockdown and GSK923295 treatment (Figure 7G,H).

## 4. Discussion

In this report we investigated the effects of CENPE knockdown and chemical inhibition on MB cells, to start assessing the potential of this target for MB therapy. For chemical inhibition, we used the GSK923295 compound, because it is already in clinical trial for other tumors [46]. Our results revealed that CENPE loss strongly limits the in vitro expansion of human MB cells, leading to high frequency of mitotic catastrophes, DNA damage and apoptosis. CENPE inhibition was more penetrant than CENPE down-regulation in inducing mitotic catastrophe and cell death.

Neural progenitors are strongly sensitive to CENPE levels, because microcephaly is a haplo-insufficient phenotype of CENPE loss [30]. Although in our study we did not compare directly MB cells with other cell types, the results support the possibility that MB cells may share with neural progenitors a particularly high sensitivity to CENPE loss. Our results are also consistent with the finding that silencing of OTX2, a transcription factor required for CENPE expression in MB cells, lead to inhibition of MB cell proliferation [59,61].

ONS-76 and DAOY were efficiently inhibited by low-nanomolar concentrations of GSK92329, with an EC50 at 72 h of 14 and 12 nM, respectively. These values are notably low, when compared to the values obtained in other cell lines. For example, in a panel of 19 human neuroblastoma derived cell lines, GSK923295 EC50 ranged between 27 and 266 nM [62]. EC50 was 250 nM in a colorectal carcinoma cell line [63], 790 nM in a cisplatin-resistant ovarian cancer cell line [64], 140 nM in a sarcoma cell line [65]. GSK923295 inhibitory activity was also tested across 237 tumor cell lines after 72 h of continuous exposure. The EC50 values spanned from 12 nM to greater than 10 µM, with an average of 253 nM and median of 32 nM [50]. Interestingly, the latter study found that the Group 3 human MB cell line D283 is also very sensitive, with an EC50 of 28 nM [50].

Based on such comparisons, it will be important to address the sensitivity of other MB cell lines, in vitro and in vivo. The latter studies will be also critical to assess whether GSK923295 can be delivered in vivo to MB cells, since it has not been established whether the compound can cross the blood brain barrier [52].

Our observations raise the question of why the tested cell lines display such high sensitivity. Based on studies performed in HeLa cells, CENPE inactivation was thought to induce mitotic catastrophe, DNA damage and cell death, but only in combination with inactivation of the SAC. On this basis, one possibility to explain the high effectiveness of CENPE knockdown and inhibition to induce these phenotypes could be that MB cells have a sub-efficient SAC. However, this seems an unlikely scenario. Indeed, while SAC genes are frequently mutated in various solid tumors including colon, breast, prostate and lung [66,67,68,69,70,71,72], a survey of four MB datasets included in the cBioPortal [4,73,74,75] revealed a lower-than-expected mutation rate in BUB1, BUB1B, BUB3, MAD1L1, MAD2L1, CDC20 (the maximal rate of inactivating mutations was 0.3%, as in the case on BUB1). A similarly low mutation frequency (0,7%) was found for CENPE, which is known as a functional component of the SAC [76]. This observation strongly suggests that MB cells have an active SAC and that this cellular function is critical for their efficient expansion. Conversely, it suggests that activation of the SAC could be a critical passage of the anti-tumor activity elicited in MB cells by CENPE inactivation. This possibility would be consistent with the fact that inhibition of SAC has been proposed for MB treatment [77].

In normal neural progenitors, mitotic delay is sufficient to induce genotoxic stress, TP53 activation and apoptosis [78]. This mechanism could well explain the phenotypes observed in ONS-76 cells, which are TP53-proficient and showed TP53 and P21 activation. Most interestingly, the anti-proliferative and pro-apoptotic effects elicited by CENPE inactivation were even more pronounced in DAOY cells, which contain a mutated TP53. As it has already been described in other TP53-deficient contexts [79,80,81,82,83], the induction of TP73 observed in DAOY cells could play a role in the response to genotoxic stress, by activating target genes such as P21 in a P53-independent manner. This is consistent with previous works that demonstrated that TP73 plays important roles in the maintenance of genomic stability [84,85,86,87,88], DNA damage repair [89,90] and in triggering apoptosis [91,92,93,94,95].

## 5. Conclusions

We have observed in this study a potent single agent anti-cancer activity of GSK923295 in two different medulloblastoma cell lines. Given the devastating and life-long side-effects of brain irradiation, especially in pediatric age groups, GSK923295 in conjunction with lower doses of irradiation and chemotherapy could provide an important alternative therapeutic strategy that is much needed for the treatment of MB.

## Figures and Tables

**Figure 1 cancers-13-01028-f001:**
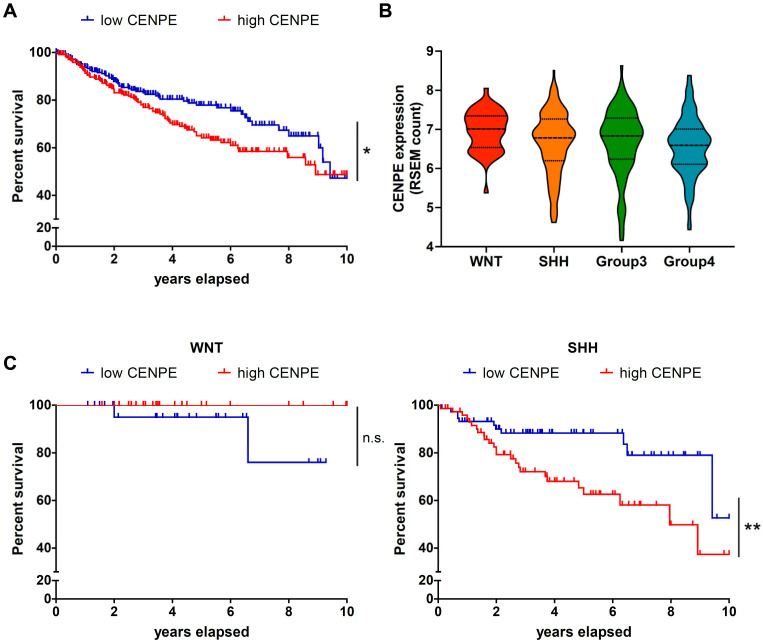
The expression of CENPE correlates with MB patients’ prognosis. (**A**) Kaplan-Meier survival curves of MB patients. The two groups were defined on the basis of CENPE expression high (red) or low (blue) based on median value. *p*-value = 0.0318, Log-rank test statistics (**B**) Graph showing expression level by RSEM count of CENPE divided by molecular subgroups (**C**) Kaplan-Meier survival curves of MB patients divide by molecular subgroups and finally subdivided in two groups on the basis of CENPE expression high (red) or low (blue) based on median value. WNT *p*-value = 0.1385 SHH *p*-value = 0.0091 Group3 *p*-value = 0.2815 Group4 *p*-value = 0.1363, Log-rank test statistics. *, *p* < 0.05; **, *p* < 0.01; n.s., not significant.

**Figure 2 cancers-13-01028-f002:**
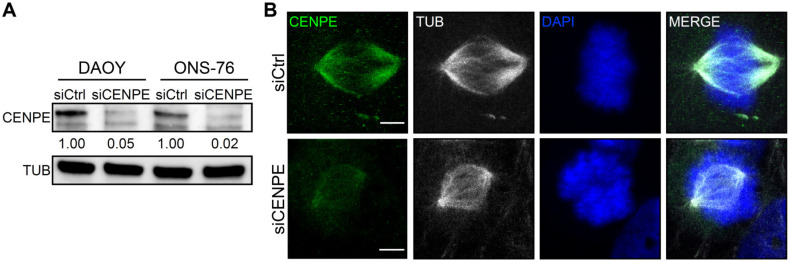
Medulloblastoma cells are sensitive to CENPE knockdown. (**A**) Western blot analysis of total lysate from DAOY and ONS-76 cell lines, 48 h after treatment with non-targeting (siCtrl) or CENPE-specific (siCENPE) siRNA. The level of CENPE was analyzed and the internal loading control was tubulin (TUB). The original images are shown in the Appendix A. (**B**) Representative image of DAOY cells processed for immunofluorescence 48 h after transfection with siCtrl or siCENPE and stained with anti-CENPE antibody, anti-Tubulin antibody and DAPI. (**C**) Representative images of live imaging performed on DAOY 30 h after transfection. Time lapses were recorded overnight with an interval of 5 min. Magnification: 40× (**D**) Quantification of the time spent in metaphase by DAOY and ONS-76 cells analyzed as described in panel C. (**E**) Quantification of the percentage of cells showing metaphases lasting more than 30 min, in cells analyzed as described in panel C. Statistical test = Chi^2^, considering absolute number of cells in experiments. (**F**,**G**) DAOY and ONS-76 proliferation assay: 50,000 cells were transfected with non-targeting (siCtrl) or CENPE-specific (siCENPE) siRNA. Growth curves were obtained by assessing cells’ number in each well at 48, 72 and 96 h after transfection. Statistical test = two-tailed Student *t*-test. All quantifications were based on at least 3 independent biological replicates. At least 60 cell divisions were recorded for any biological replicate. Error bars, SEM. *, *p* < 0.05; **, *p* < 0.01; ***, *p* < 0.001. Scale bars, 5 µm.

**Figure 3 cancers-13-01028-f003:**
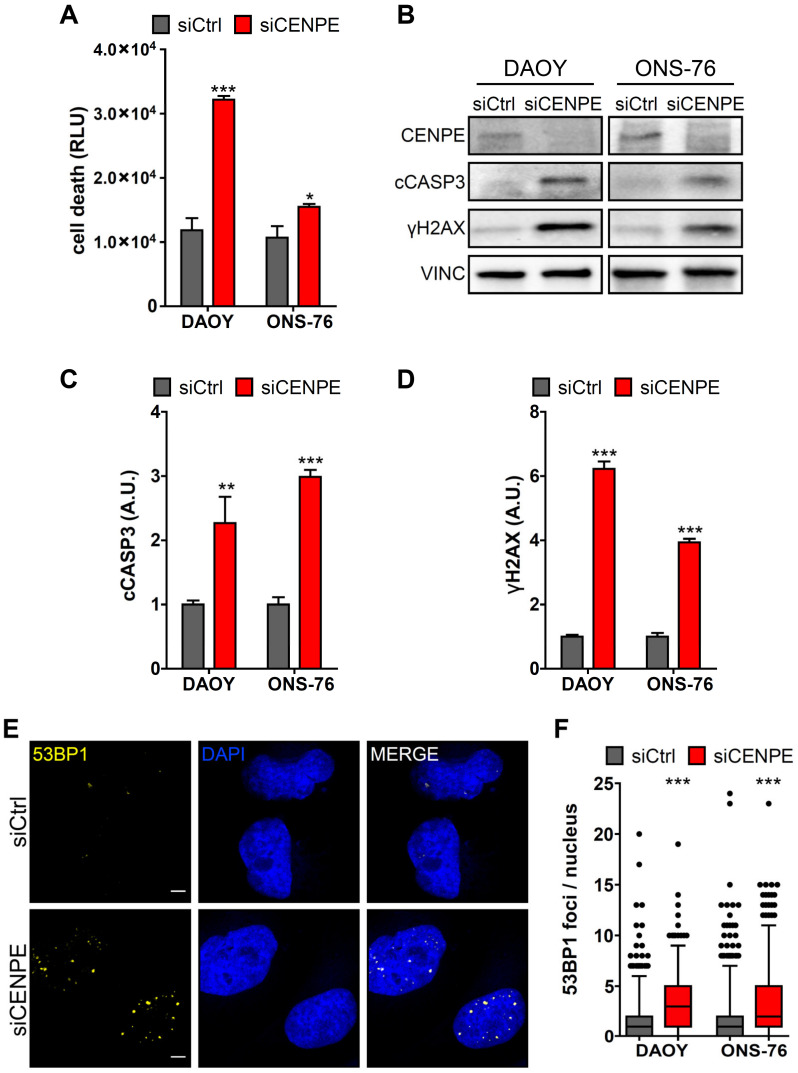
CENPE knockdown induces apoptosis and DNA damage in MB cells. (**A**) DAOY and ONS-76 cell death, measured by CellTox™ Green Cytotoxicity assay, 48 h after treatment with siCtrl or siCENPE. Values are plotted as Relative Light Unit (RLU). (**B**) Western blot analysis of total lysate from DAOY and ONS-75 cell lines, 48 h after treatment with siCtrl or siCENPE. The levels of CENPE, cleaved Caspase 3 (cCASP3) and γH2AX were analyzed and the internal loading control was vinculin (VINC). The original images are shown in the Appendix A. (**C,D**) Quantification of the relative density of cCASP3 (C) and γH2AX in DAOY and ONS-76 treated cells. (**E**) Representative image of DAOY cells processed for immunofluorescence 48 h after transfection with siCtrl or siCENPE and stained with anti-53BP1 antibody and DAPI. (**F**) Quantification of 53BP1 foci per nucleus in cells treated as in (**E**). All quantifications were based on at least three independent biological replicates. 53BP1 foci were counted in >300 cells per condition. Error bars, SEM. *, *p* < 0.05, **, *p* < 0.01; two-tailed Student *t*-test for blots and CellTox™ assay. ***, *p* < 0.001 Mann–Withney U test for 53BP1 foci. Scale bars, 5 µm. A.U., arbitrary units.

**Figure 4 cancers-13-01028-f004:**
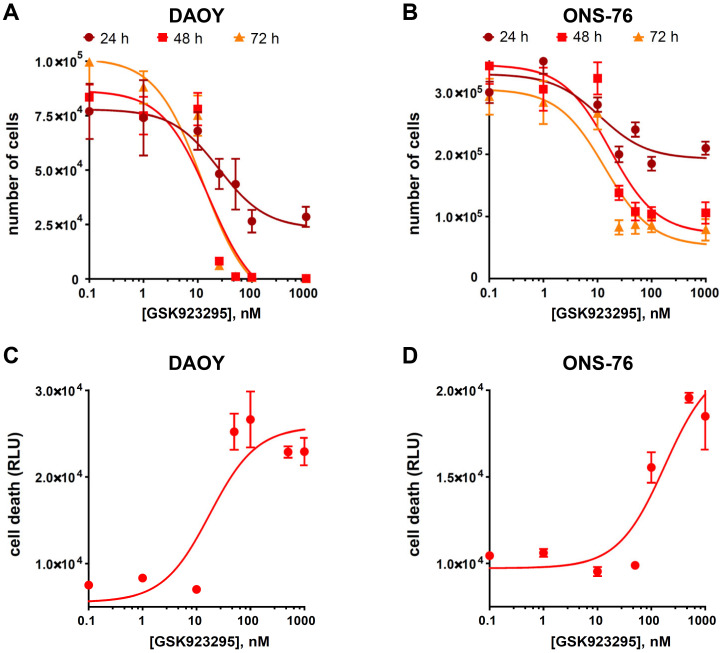
CENPE inhibition reduces cell proliferation and induces cell death. (**A**,**B**) Dose response curve of DAOY and ONS-76 cells, respectively, 24, 48 and 72 h after treatment with GSK923295. (**C**,**D**) DAOY and ONS-76 cell death, measured by CellTox™ Green Cytotoxicity assay, 24 h after treatment with DMSO or GSK923295. Values are plotted as Relative Light Unit (RLU). (**E**,**F**) DAOY and ONS-76 proliferation assay: 50,000 cells were treated with DMSO or 25 nM GSK923295. Growth curves were obtained by assessing cells’ number in each well 24, 48 and 72 h after treatment. All quantifications were based on at least three independent biological replicates. Error bars, SEM; *, *p* < 0.05; ***, *p* < 0.001; two-tailed Student *t*-test.

**Figure 5 cancers-13-01028-f005:**
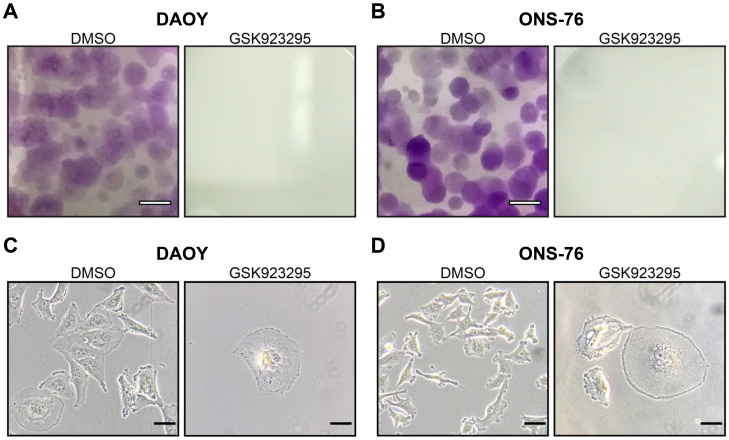
CENPE inhibition abolished the in vitro clonogenic potential of SHH MB cells. (**A**,**B**) Low magnification images of plates of DAOY and ONS-76 cells, treated with DMSO or GSK923295 25 nM for 7 days, stained with crystal violet. Scale bars 25 mm (**C**,**D**) Example of DAOY and ONS-76 treated with DMSO or GSK923295 25 nM for 7 days in phase contrast. Scale bar 20 μm.

**Figure 6 cancers-13-01028-f006:**
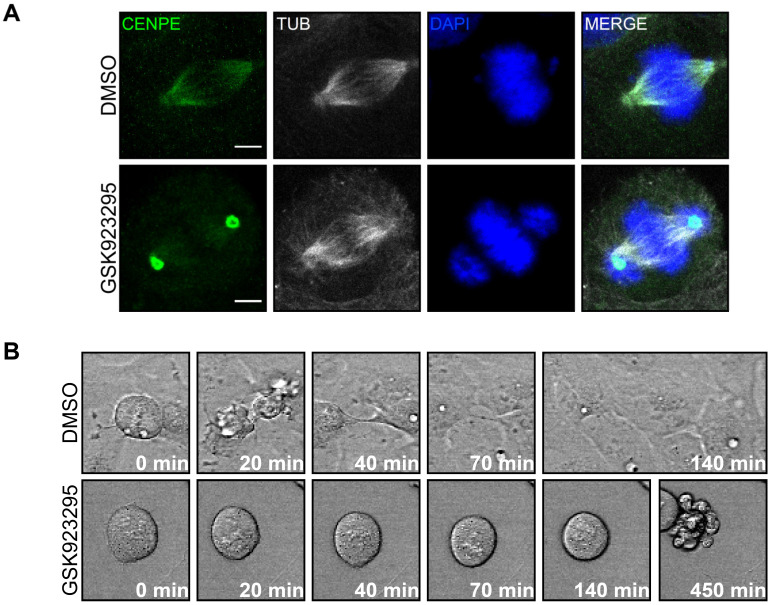
CENPE inhibition alters mitotic spindle assembly and induces mitotic catastrophe. (**A**) Representative images of DAOY cells, processed for immunofluorescence 24 h after treatment with DMSO or 25 nM GSK923295, immunostained with anti-CENPE and anti-Tubulin antibodies and counterstained with DAPI. (**B**) Representative frames of live-imaging experiments, performed on DAOY cells 24 h after treatment with DMSO or 25 nM GSK923295. Time lapses were recorded overnight with an interval of 5 min. (**C**) Quantification of the time spent in metaphase, in DAOY and ONS-76 cells analyzed as described in panel B. Statistical test = two-tailed Student *t* test. (**D**) Quantification of the percentage of metaphases lasting more than 30 min, in DAOY and ONS-76 cells analyzed as described in panel B. Statistical test = Chi^2^, considering absolute number of cells in experiments (**E**) Quantification of the percentage of correct cell division (Division), cytokinesis failure (Failure) and mitotic catastrophe (Death) in DAOY and ONS-76 cells analyzed as described in panel B. Statistical test = Chi^2^, considering absolute number of cells in experiments (**F**) Western blot analysis of total lysates from DAOY and ONS-75 cells, 24 h after treatment with DMSO or 25 nM GSK923295. The levels of CENPE, cleaved Caspase 3 (cCASP3) and γ H2AX were analyzed and the internal loading control was vinculin (VINC). The original images are shown in the Appendix A (**G**,**H**) Quantification of the relative density of cCASP3 (**G**) and γH2AX (H) in DAOY and ONS-76 cells treated as described in panel A. Statistical test = two-tailed Student *t*-test. All quantifications were based on at least three independent biological replicates. At least 60 cell divisions were recorded for any biological replicate. Error bars, SEM; **, *p* < 0.01; ***, *p* < 0.001. Scale bars, 5 µm.

**Figure 7 cancers-13-01028-f007:**
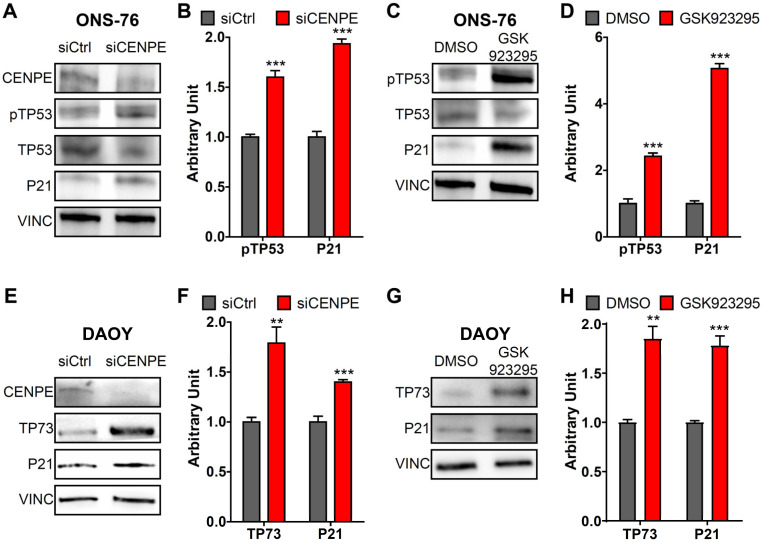
CENPE depletion or inhibition induces TP53 or TP73 pathway. (**A**,**B**) Western blot analysis and quantification of total lysate from ONS-76, 48 h after treatment with the indicated siRNAs. The levels of phospho-Ser15-TP53 (pTP53) and P21 were analyzed, the internal loading control was vinculin (VINC). (**C**,**D**) Western blot analysis and quantification, performed as in panel A, of total lysate from ONS-76, 24 h after treatment with 25 nM GSK923295. (**E**,**F**) Western blot analysis and quantification of total lysate from DAOY, 48 h after treatment with the indicated siRNAs. The levels of TP73 and P21 were analyzed, the internal loading control was vinculin (VINC). (**G**,**H**) Western blot analysis and quantification, performed as in panel E, of total lysate from DAOY, 24 h after treatment with 25 nM GSK923295. All quantifications were based on at least four independent biological replicates. Error bars, SEM; **, *p* < 0.01; ***, *p* < 0.001; two-tailed Student *t*-test. The original images are shown in the Appendix A.

## Data Availability

Kaplan-Meier survival analysis and CENPE expression data presented in this study are openly available in GEO under the accession number GSE85218.

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
