# Peer review of "CENPE Inhibition Leads to Mitotic Catastrophe and DNA Damage in Medulloblastoma Cells"

_cancers, 2021, doi:10.3390/cancers13051028_

Round 1
Reviewer 1 Report
In this study Iegiani et al. addressed the role CENPE, a centrosome-associated protein, in medulloblastoma. Since mutations in CENPE cause primary microcephaly syndrome by affecting neural progenitors, a proposed cell of origin of medulloblastoma, the authors hypothesize that CENPE could influence tumor growth. Indeed, knockdown and pharmacological inhibition of CENPE, drastically reduce cell growth of two medulloblastoma cell lines by interfering with their cell division. Although the study is conducted in vitro and limited to only a specific aspect of CENPE inhibition, it is well done and it investigates a novel promising target for a disease with few effective therapeutic options.
Here are the specific issues that need to be addressed:
Major points:
- Is CENPE overexpressed in medulloblastoma? Since, as the authors pointed out, no data on the role of CENPE are available for medulloblastoma, they should provide evidence that blocking CENPE could be a promising approach, worth of further investigation in vivo. The authors are requested to show whether CENPE is upregulated in medulloblastoma in vivo and not only in cell lines. The authors can use tissues from mouse models or human samples or, in alternative, use publicly available datasets. Analysis of medulloblastoma subgroups is also appreciated and could provide an additional rational to the use of cell lines belonging to the Shh-subgroup.
- Although CENPE knockdown or pharmacological inhibition is unquestionably efficient in blocking cell proliferation and inducing cell death, there is always a proportion of cells that resist to the treatment. Are these resistant cells still able to proliferate or do they undergo senescence or differentiation? This is particular important for ONS-76 cells that seem to respond less to the treatments.
- In the WB analysis of phosphorylated TP53 the authors should include also a WB of total TP53.
Minor points:
- Can GSK923295 pass the blood-brain barrier? Since the authors suggest this drug as a possible new treatment for brain tumors, they should mention in the introduction or discussion if studies regarding its brain permeability are available.
- It is not clear how the authors used a Chi-squared test for data in figure 1E and 4 C,D. It would be correct to use the Chi-squared test if the authors have analysed absolute frequency (i.e. number of cells, not percentages) and have considered the distribution of cells with metaphase >30 minutes and <30 minutes between siCtrl and siCENPE. If that is the case the authors can leave the graph showing only % of metaphases >30 minutes but they should clarify in the legends or Materials and Methods on which type of data they have performed the Chi-squared test.
- In figure 3A-B the error bars are missing for all the data points. Are they not visible because too small or are these experiments performed only once?
- In Figure 5H the y axis is missing
- Please, give more details of the Colony Forming Assay in the Materials and Methods sections.
- Please, define in the Materials and Methods section how the live imaging videos have been analysed. Which software has been used? How did the authors define a metaphase, a mitosis failure and a mitosis catastrophe? Did they manually identify such cell states or did they use an automatic method with software/plug-in?
Author Response
In this study Iegiani et al. addressed the role CENPE, a centrosome-associated protein, in medulloblastoma. Since mutations in CENPE cause primary microcephaly syndrome by affecting neural progenitors, a proposed cell of origin of medulloblastoma, the authors hypothesize that CENPE could influence tumor growth. Indeed, knockdown and pharmacological inhibition of CENPE, drastically reduce cell growth of two medulloblastoma cell lines by interfering with their cell division. Although the study is conducted in vitro and limited to only a specific aspect of CENPE inhibition, it is well done and it investigates a novel promising target for a disease with few effective therapeutic options.
We thank the Referee for her/his generally positive evaluation of our work and for the constructive criticisms.
Here are the specific issues that need to be addressed:
Major points:
- Is CENPE overexpressed in medulloblastoma? Since, as the authors pointed out, no data on the role of CENPE are available for medulloblastoma, they should provide evidence that blocking CENPE could be a promising approach, worth of further investigation in vivo. The authors are requested to show whether CENPE is upregulated in medulloblastoma in vivo and not only in cell lines. The authors can use tissues from mouse models or human samples or, in alternative, use publicly available datasets. Analysis of medulloblastoma subgroups is also appreciated and could provide an additional rational to the use of cell lines belonging to the Shh-subgroup.
We thank the Referee for this suggestion. We have integrated the revised manuscript with an analysis of MB patients’ data contained in a published study (Cavalli et al., 2017), including 469 MB from different subgroups. As reported in the text, low expression of CENPE correlates in general with a better prognosis. Considering the different subgroups, no statistically significant differences were noted in average CENPE expression levels, but low expression significantly correlates with better prognosis in the SHH subtype, compared to other subgroups. This result further justifies, a posteriori, our choice to concentrate our efforts on this subtype. For this reason, we have decided to include the new data at the beginning of the results section.
- Although CENPE knockdown or pharmacological inhibition is unquestionably efficient in blocking cell proliferation and inducing cell death, there is always a proportion of cells that resist to the treatment. Are these resistant cells still able to proliferate or do they undergo senescence or differentiation? This is particular important for ONS-76 cells that seem to respond less to the treatments.
To address this question, we plated 10000 cells in six well plates to analyze the phenotype of cells surviving after the GSK923295 treatment. As reported in the text:” In every plate, single cells or abortive colonies (less than 10 cell per colony) were detected. Among them, 86% and 73%, respectively, were composed of single cells, characterized by a flattened senescent morphology”
- In the WB analysis of phosphorylated TP53 the authors should include also a WB of total TP53.
We enclosed total TP53 westernblot in the figure (now figure 7).
Minor points:
- Can GSK923295 pass the blood-brain barrier? Since the authors suggest this drug as a possible new treatment for brain tumors, they should mention in the introduction or discussion if studies regarding its brain permeability are available.
To our knowledge, no data are so far available on GSK923295 brain permeability. We think this will be an important point to elucidate in future experiments, to promote GSK923295 as MB treatment. We have added this statement to the discussion with this statement
- It is not clear how the authors used a Chi-squared test for data in figure 1E and 4 C,D. It would be correct to use the Chi-squared test if the authors have analysed absolute frequency (i.e. number of cells, not percentages) and have considered the distribution of cells with metaphase >30 minutes and <30 minutes between siCtrl and siCENPE. If that is the case the authors can leave the graph showing only % of metaphases >30 minutes but they should clarify in the legends or Materials and Methods on which type of data they have performed the Chi-squared test.
We analyzed absolute frequency of the data. In revised manuscript, we specified in M&M and in figure legend (now figure 2 and 6)
- In figure 3A-B the error bars are missing for all the data points. Are they not visible because too small or are these experiments performed only once?
Error bars were omitted to make graph more clear. In revised manuscript, we integrated bar errors in the graph (now figure4)
- In Figure 5H the y axis is missing
We completed the figure (now figure7)
- Please, give more details of the Colony Forming Assay in the Materials and Methods sections.
We integrated M&M with more details
- Please, define in the Materials and Methods section how the live imaging videos have been analysed. Which software has been used? How did the authors define a metaphase, a mitosis failure and a mitosis catastrophe? Did they manually identify such cell states or did they use an automatic method with software/plug-in?
We integrated M&M with requested details
Reviewer 2 Report
In this paper by Iegiani et al., the aim of the authors was to highlight more effective therapies than the current applied protocol in the treatment of medulloblastoma. They looked for mutated genes in primary microcephaly that have been also proposed as possible targets for medulloblastoma-directed drug development. They focused their attention on Centromere-associated protein E (CENPE), which is important in chromosome congression, spindle microtubule capture at kinetochores and spindle assembly checkpoint. Using several in vitro approaches, they showed that downregulation of CENPE expression using siRNA or its inhibition using a specific inhibitor impaired the growth of two medulloblastoma cell lines. They concluded that CENPE could be a promising target for medulloblastoma treatment.
The paper is well written and presented and the undertaken experiments are solid. However, in its current state, this study lacks originality and does not provide new insights to the field. Moreover, some statements of the authors are not clear enough. Please find below my specific concerns:
- As stated by the authors, CENPE activity is necessary not only for cancer cells, but also for normal cells, such as neural progenitors. Taking into account the importance of CENPE during cell cycle, it is obvious that impairing its activity will have toxic effects on any cell type. The authors did not include any normal cell line (neural stem or progenitor cells) in their analysis as a reference point. This step is important if authors want to conclude that targeting CENPE will specifically impair medulloblastoma cells. It is important to find if there are concentrations of the CENPE inhibitor that affect cancer cells while modestly affecting normal cells. Ideally authors should include normal cells with varying proliferation profile and test if only the highly proliferating cells are affected. Furthermore, could the authors explain why they used 25nM of GSK923295 in their long-term proliferation assay?
- It is frequent to find that deregulation of the same genes or gene network is implicated in different pathological contexts. In this study, it is not clear to me what is the rationale in focusing on genes implicated in primary microcephaly in order to identify robust targets for medulloblastoma treatments. The authors should clarify this point in order to justify their choice.
- Balamuth et al. (2010), already focused their attention on CENPE as a potential target in medulloblastoma. In this study, the authors treated 19 human neuroblastoma-derived cell lines with GSK923295, and evaluated the consequences of this treatment in vitro and in vivo. While the authors cited this paper in order to compare the range of concentrations they used for CENPE inhibitor, they did not discuss and compare their results to the one of Balamuth et al. Authors should highlight whether their results are coherent or not with the one of this paper. Moreover, other reports have highlighted the importance of CENPE in medulloblastoma biology. Bunt et al, identified that CENPE acts downstream of the oncogene OTX2. All these data should be discussed by the authors in order to further support their results and the importance to focus on this protein.
- Lately, several dataset of human medulloblastoma transcriptomes are being available. These datasets are either generated at the bulk or at the single cell level. They provide the closest view of the tumor reality at play within patients. Authors should investigate the clinical relevance of their findings by analyzing CENPE expression in these datasets. This step should enhance the originality of the paper and provide new findings.
Some minor concerns:
- Please explain the ATCC protocol you used.
- Panel 1D should indicate what does it show (time spent in meta-phase). Panel 1E should add « % of cells… » in the Y axis title. Panel 2A should indicate what does it show (Death RLU). Panel 3C-D should indicate what does it show (Death RLU).
- Please check for some minor English errors
Author Response
In this paper by Iegiani et al., the aim of the authors was to highlight more effective therapies than the current applied protocol in the treatment of medulloblastoma. They looked for mutated genes in primary microcephaly that have been also proposed as possible targets for medulloblastoma-directed drug development. They focused their attention on Centromere-associated protein E (CENPE), which is important in chromosome congression, spindle microtubule capture at kinetochores and spindle assembly checkpoint. Using several in vitro approaches, they showed that downregulation of CENPE expression using siRNA or its inhibition using a specific inhibitor impaired the growth of two medulloblastoma cell lines. They concluded that CENPE could be a promising target for medulloblastoma treatment.
The paper is well written and presented and the undertaken experiments are solid. However, in its current state, this study lacks originality and does not provide new insights to the field. Moreover, some statements of the authors are not clear enough.
We thank the Referee for her/his generally positive evaluation of our work and for the constructive criticisms.
Please find below my specific concerns:
- As stated by the authors, CENPE activity is necessary not only for cancer cells, but also for normal cells, such as neural progenitors. Taking into account the importance of CENPE during cell cycle, it is obvious that impairing its activity will have toxic effects on any cell type. The authors did not include any normal cell line (neural stem or progenitor cells) in their analysis as a reference point. This step is important if authors want to conclude that targeting CENPE will specifically impair medulloblastoma cells. It is important to find if there are concentrations of the CENPE inhibitor that affect cancer cells while modestly affecting normal cells. Ideally authors should include normal cells with varying proliferation profile and test if only the highly proliferating cells are affected.
In principle, we agree with the Referee. In practice, we believe that carefully addressing this point would be a very demanding task. In general, we would argue that most microcephaly genes, although playing clear roles in cell division and genome stability, are strongly required only in few cell types, especially in CNS progenitors. The situation of CENPE is even more complex. In this case, the idea that CNS progenitors are much more sensitive than other cell types to protein function has strong genetic support, because the microcephaly phenotype produced by CENPE null mutations is haplo-insufficient. This implies that most cell types are OK with 50% of normal protein levels, while neural progenitors are severely affected. Differently from other microcephaly genes, total inactivation of CENPE is embryonically lethal. On this basis, we would expect a graded sensitivity of normal cell types to CENPE levels and/or function, which must be addressed in detail by pharmacological studies.
A comparison between normal CNS progenitors and MB cells would be sensible, but not very relevant therapeutically, because in most cases tumors arise after completion of CNS development. A comparison between tumors and differentiated neural cells would not provide relevant information since, as noted by the Referee, CENPE is well expressed only in proliferating cells. For these reasons, we limited ourselves at comparing in the discussion the EC50 reported for many different types of tumor cell lines. This comparison supports the conclusion that MB cells are characterized by a relatively low EC50. The only way to really address the point raised by the Referee would be to use MB in vivo models and compare the sensitivity of tumor cells to the different proliferating normal cell types. However, we believe that such an approach would go well beyond the aim of the present study.
Furthermore, could the authors explain why they used 25nM of GSK923295 in their long-term proliferation assay?
We used 25nM for long proliferation assay and molecular analysis because it is the lower effective dose on both cell lines at 24h post treatment. We specified this point in the text of the revised manuscript.
- It is frequent to find that deregulation of the same genes or gene network is implicated in different pathological contexts. In this study, it is not clear to me what is the rationale in focusing on genes implicated in primary microcephaly in order to identify robust targets for medulloblastoma treatments. The authors should clarify this point in order to justify their choice.
As noted above, although MCPH gene are expressed by all proliferative cells, they are specifically required in CNS progenitors. The manuscripts rationale is that MB originates and shares many molecular profiles with neural progenitors, which are is the most affected cell types in MCPH syndromes. Therefore, a similar sensitivity could be expected in MB, as well as in other tumors of neural origin. The high sensitivity to CENPE levels is further supported by the fact that, in MB datasets, CENPE as well as SAC genes show less inactivating mutations than expected by chance, as we reported in the discussion. In the revised manuscript we have tried to make more clear these points, especially in the introduction.
- Balamuth et al. (2010), already focused their attention on CENPE as a potential target in medulloblastoma. In this study, the authors treated 19 human neuroblastoma-derived cell lines with GSK923295, and evaluated the consequences of this treatment in vitro and in vivo. While the authors cited this paper in order to compare the range of concentrations they used for CENPE inhibitor, they did not discuss and compare their results to the one of Balamuth et al. Authors should highlight whether their results are coherent or not with the one of this paper.
As noted by the Referee, the study by Balamuth et al. is focused on Neuroblastoma, more than on MB. However, as we have underscored in the previous point, based on our starting hypothesis, we would expect a high sensitivity even in neuroblastoma. For this reason, we already included the results of Balamuth et al. in our discussion of the relative sensitivity of different tumor cell lines to GSK923295.
Moreover, other reports have highlighted the importance of CENPE in medulloblastoma biology. Bunt et al, identified that CENPE acts downstream of the oncogene OTX2. All these data should be discussed by the authors in order to further support their results and the importance to focus on this protein
As above, we agree with the Referee and have now included the results obtained in Bunt et al. (2010 and 2012)
- Lately, several dataset of human medulloblastoma transcriptomes are being available. These datasets are either generated at the bulk or at the single cell level. They provide the closest view of the tumor reality at play within patients. Authors should investigate the clinical relevance of their findings by analyzing CENPE expression in these datasets. This step should enhance the originality of the paper and provide new findings.
We thank the Referee for this suggestion. We have integrated the revised manuscript with an analysis of MB patients’ data contained in a published study (Cavalli et al., 2017), including 469 MB from different subgroups. As reported in the text, low expression of CENPE correlates in general with a better prognosis. Considering the different subgroups, no statistically significant differences were noted in average CENPE expression levels, but low expression significantly correlates with better prognosis in the SHH subtype, compared to other subgroups. This result further justifies, a posteriori, our choice to concentrate our efforts on this subtype. For this reason, we have decided to include the new data at the beginning of the results section.
Some minor concerns:
- Please explain the ATCC protocol you used.
We have added this detail in the revised manuscript
- Panel 1D should indicate what does it show (time spent in meta-phase). Panel 1E should add « % of cells… » in the Y axis title. Panel 2A should indicate what does it show (Death RLU). Panel 3C-D should indicate what does it show (Death RLU).
We thank the Referee for the suggestions and corrected the figures accordingly
- Please check for some minor English errors
We made our best with text-editing.
Round 2
Reviewer 1 Report
The authors have answered all my questions and modified the manuscript accordingly. I'm now satisfied by the revised form of the manuscript.
I noticed just a small error in the Discussion: the sentence in line 385-387 seems to be missing something. Possibly "blood brain barrier" at the end of the sentence?
Reviewer 2 Report
Authors replied to my concerns. I have no more comments.